# Anti-SARS-CoV-2 Activity of Surgical Masks Infused with Quaternary Ammonium Salts

**DOI:** 10.3390/v13060960

**Published:** 2021-05-22

**Authors:** Gary S. Selwyn, Chunyan Ye, Steven B. Bradfute

**Affiliations:** 1Green Theme Technologies Inc., Rio Rancho, NM 87124, USA; 2Center for Global Health, Department of Internal Medicine, University of New Mexico Health Sciences Center, Albuquerque, NM 87131 USA; cye@salud.unm.edu

**Keywords:** SARS-CoV-2, inactivation, personal protective equipment, masks

## Abstract

The SARS-CoV-2 pandemic has highlighted the need for protective and effective personal protective equipment (PPE). Research has shown that SARS-CoV-2 can survive on personal protective equipment, such as commonly used surgical masks. Methods are needed to inactivate virus on contaminated material. We show here that embedding viral-disinfecting compounds during the manufacturing of surgical masks inactivates a high dose (up to 1 × 10^5^ pfu) of live, authentic SARS-CoV-2 within minutes.

## 1. Introduction

SARS-CoV-2 has caused a devastating pandemic that is still ongoing. Public health measures were undertaken to slow the spread of the virus, including a heavy focus on social distancing and the use of personal protective equipment (PPE) to decrease rates of infection. The most common form of PPE used by the general public has been masks, including common disposable surgical masks. However, it has been well-documented that infectious SARS-CoV-2 can live on surfaces and masks for 1–28 days, depending on the material tested [1,2,3], raising concerns over potential infection by improperly handled contaminated PPE. Furthermore, sharing or re-use of PPE in resource-constrained locations (or situations in which there is a supply shortage of PPE) could lead to increased infection with SARS-CoV-2 or other respiratory viruses. One way to address this concern for SARS-CoV-2 and other viruses is to embed antiviral substances in PPE to inactivate any contaminating virus. This could lead to reduced viral spread and longer lasting PPE, which could alleviate shortages in the PPE supply chain. In this study, we analyzed the efficacy of antiviral substances embedded into surgical masks for activity against live SARS-CoV-2. Our studies indicated that complete inactivation of high titers of live SARS-CoV-2 (an inoculum of 1 × 10^5^ pfu) was achieved on treated surgical masks. These data suggest that PPE can be protected against viral contamination, which has implications for increased safety and life span of common PPE materials.

## 2. Materials and Methods

All surgical masks tested (FILTR brand surgical masks, an approved Class I medical device) had a spunbond polypropylene (PP) outer fabric, an electrostatically charged meltblown PP middle filtration layer, and an inner spunbond PP fabric that contacts the face. A mixture of disinfectant chemicals were embedded into a polymer film (265L, a free-radical, acrylate-type polymer film that is thermally stable and fluorocarbon free, commercially produced by Green Theme Technologies, following U.S. Patent 10,655,272) applied to the outside (spunbond PP layer) of the surgical mask. Several different disinfectant chemistries were studied and evaluated: (1) phenolics (2-phenylphenol and thymol), (2) quaternary ammonium compounds of different compositions and water solubility, and (3) the natural products peppermint oil and citric acid. Monomer chemistry was selected to dissolve the disinfectant chemicals and was coated onto only the outside spunbond layer of the masks by gravure coating at 2–3 g per square meter. The coated film was polymerized by heating to 100 °C in a pressure vessel with N_2_ at 3450 KPa. This produced a microscopic, invisible polymer film with the embedded disinfectants or natural products bonded to the outside of the mask. The heat and pressure sterilized the masks, and they were subsequently wrapped in sterile aluminum foil. Masks were cut in small pieces to provide 2–3 samples for each chemical treatment, with two samples in the single-experiment screens in Figure 1 and three samples in two separate tests for the experiments in Figure 2. Mask pieces were inoculated with approximately 1 × 10^5^ plaque forming units (pfu) of live SARS-CoV2 virus (isolate USA-WA1/2020, from BEI Resources) in a BSL-3 laboratory for the indicated times. The virus stocks were amplified in Vero E6 cells in DMEM/5% FCS and supernatants were harvested and quantitated. Mask pieces were inoculated with 12.5 µL of virus. The inoculated fabric was incubated for the indicated times, and the virus was extracted by incubating mask pieces with 300 µL of cell culture medium for 30 min. The eluted virus was added at a 1:10 dilution to Dey-Engley Neutralizing Broth (Sigma Aldrich, St Louis, MO, USA) for 15 min at room temperature, and then the live virus was quantitated using a plaque assay on Vero E6 cells as we have previously published [4,5,6], with a limit of detection of 1 pfu per test. The viral reduction was calculated by comparison to the control samples for the same time period.

## 3. Results

We conducted a preliminary screen of surgical masks embedded with compounds known to have antiviral activity. Masks containing a polymer (265 L) used to anchor the compounds and combinations of a water-soluble quaternary ammonium compound (labeled 3310), a water insoluble quaternary ammonium compound (labeled 1297), and the disinfectant 2-phenylphenol (with or without 10% PEG 200) completely inactivated masks inoculated with 1 × 10^5^ pfu of live, authentic SARS-CoV-2 after 2 h (Figure 1A). Masks containing combinations of the 3310 quaternary ammonium compound, thymol (a natural disinfectant), and a third water-insoluble quaternary ammonium compound, DDMAB (with or without 10% PEG200), also completely inactivated the virus. Additional combinations containing 3310, 1297, and 2-phenylphenol were similarly effective. Masks embedded with only DDMAB and thymol partially inactivated virus after 2 h (Figure 1A, 3 log10 reduction). As expected, untreated control masks or masks with only the 265L polymer did not inactivate SARS-CoV-2.

In a second screen, masks treated with additional disinfectant chemicals were inoculated with virus for 2 h (Figure 1B). In this set, thymol was substituted for 2-phenylphenol and some essential oils were similarly tested. Combinations of 1297 and various concentrations of thymol completely inactivated SARS-CoV-2. However, a combination of thymol and peppermint oil did not inactivate virus at 2 h (Figure 1B).

Based on these initial results, we conducted similar experiments with shorter time points. We found complete inactivation of SARS-CoV-2 with masks infused with a combination of 1297, 3% thymol, and citric acid after 20 or 60 min, and, in some experiments, in as few as 5 min, showing remarkable rapid potency (Figure 2A–C). Complete inactivation was achieved with 1297 alone at 60 min, with partial viral inactivation at 5 and 20 min, although we did not detect any viral inactivation with thymol alone (Figure 1B) or citric acid alone (Figure 2A–C). We also found partial viral inactivation at 5 and 20 min, and complete viral inactivation at 60 min, with the following combinations: 1% thymol plus 1297; 1% thymol plus citric acid plus 1297; and 3% thymol with 1297.

## 4. Discussion 

These data show rapid and complete inactivation of high titers of live, authentic SARS-CoV-2 by masks infused with quaternary ammonium salts. The advantage of such an approach is the elimination of infectious virus on the surface of a mask, increasing its protective capabilities and minimizing accidental spread to the wearer or to shared PPE. An interesting inadvertent finding of these experiments were that there was recoverable live virus in untreated masks at the 24-h time points (data not shown), confirming our earlier studies that this virus can survive on masks used as PPE [1]. These data highlight the need for infusion of PPE with anti-viral compounds.

Quaternary ammonium salts are known to inactivate pathogens and are routinely used to disinfect surfaces. Their mechanism of action against enveloped viruses is thought to be disruption of the viral lipid envelope [7]. There is some concern that SARS-CoV-2 could be more stable in the environment based on shell disorder analysis [8]. It is possible that combinations of antiviral substances, such as quaternary ammonium salts and phenolics, may be more effective against this virus. Indeed, we found more rapid antiviral activity with combinations of the quaternary ammonium 1297 when combined with other compounds than when used alone (Figure 2 A–C). Future studies should be conducted to identify the optimal combination of multiple compounds for inactivation of SARS-CoV-2 on masks.

To address the concern of potential toxicity to the wearer, we embedded the quaternary ammonium compounds into the polymer film that is bonded to the outside of the spunbond polypropylene fabric of the 3-layer mask. Our polymerization technology, which we also use for polymeric, water repellency finishing of apparel, consists of a hydrocarbon-based monomer with an organic peroxide polymerization initiator. Because we apply the coating only to the outside of the mask, the resultant polymeric film anchors and concentrates the disinfectant chemicals in a film applied to the outside of the mask. This polymerization anchors the chemicals in place, preventing flaking, loss, or inhalation of particles. In addition, this approach minimizes any contact risk of the quaternary ammonium salts with the skin. It is important to note that the middle and inner PP mask layers are not modified by our process. The PP layer that contacts the skin has no chemical treatment. 

Untreated PP is strongly hydrophobic, meaning that water drops will not wick into untreated PP. Similarly, viral-containing droplets may have a long lifetime on PP because this protective mechanism. The quaternary compounds used are hydrophilic and, thus, cause the polymer film to be hydrophilic. As such, water droplets will rapidly wick into the polymer film. Any virus particles will be similarly absorbed with the water and will be brought into contact with the entrapped quaternary ammonium compounds. This helps kill the virus by maximizing contact of the virus into contact with the disinfectants.

This study was aimed at demonstrating an anti-viral treatment on disposable masks, in support of an FDA certification process. As such, the re-use and longevity of the treatment was not studied. However, preliminary studies by our group have shown that compounds embedded into the 265 L polymer persist through multiple laundry cycles.

Together, these data suggest that compounds placed on PPE can be used to inactivate SARS-CoV-2 in a rapid and safe manner. Quaternary ammonium salts are used to inactivate a wide range of viruses, bacteria and fungi [9,10,11,12]. Therefore, this technology should be tested for applicability to other pathogens.

## 5. Patents

A patent application is pending on this technology (GSS).

## Figures and Tables

**Figure 1 viruses-13-00960-f001:**
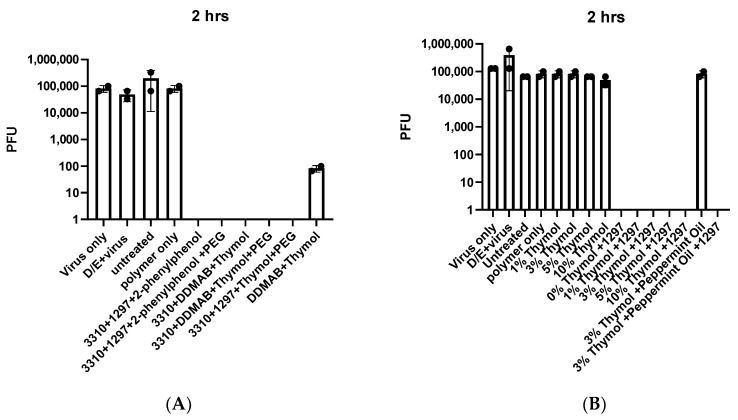
Screen for inactivation of SARS-CoV-2 by surgical masks infused with quaternary ammonium compounds. *n* = 2, separate experiments in (**A**,**B**). Error bars show standard deviation. Y axis shows total PFU detected in each sample.

**Figure 2 viruses-13-00960-f002:**
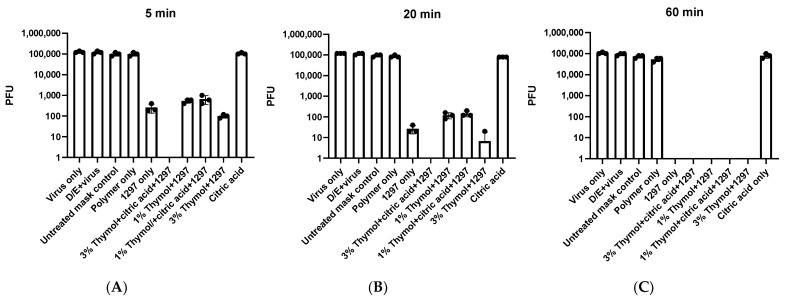
Rapid inactivation of SARS-CoV-2 by quaternary ammonium salt-infused masks. (**A**–**C**) *n* = 3, shown is representative from 2 separate experiments.

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
