# Peer review of "Anti-SARS-CoV-2 Activity of Surgical Masks Infused with Quaternary Ammonium Salts"

_viruses, 2021, doi:10.3390/v13060960_

Round 1

Reviewer 1 Report

The testing of virucidal activity of decontamination methods against SARS-CoV-2 is timely and, here, even more with a virucidal pretreatment of masks. This short paper addresses these needs. It investigates the inactivation capacity of the coating of surgical masks with quaternary ammonium compounds. A virucidal effect is evidenced against SARS-CoV-2 and apparently there is no doubt about this result. However, acceptance of this article should be subject to a more precise description of the methods and a correct presentation of the results, as explained in the following comments. Indeed an elementary rule of a primary scientific publication is that results can be reproduced. This paper does not give enough information pertaining the reproduction of the work by another laboratory:

  • Abstract: SARS-CoV-2 instead of “SARs-coV-2”
  • Abstract: “inactivates a high dose of …”: please be more precise in this statement even if it is in the summary
  • Line24: “for days”: be more precise
  • Line33: “high titers of live SARS-CoV-2”: be more precise about SARS-CoV-2 titres
  • Line37: surgical masks: which brand? Is it a regular officially approved surgical mask?
  • Line 39: what is the composition of the disinfectant chemical embedded on the outside mask layer? (There is some information in the result section but it should appear in this section)
  • Line46: how many mask pieces have been used? How many replicates in this trial? (Some information can be found in the figures but it needs to be stated in this section)
  • Line46: How masks have been inoculated? Deposit of droplets? Which volume, etc. In which medium was the virus resuspended?
  • Line46: provide a reference regarding the pfu method and the cell line used (Vero E6 cells)
  • Line 48: give the indicated times (even if they are found somewhere in the results)
  • Line49: extraction simply by incubation, without any agitation?
  • Viral reduction? Comparison is not a result: it should be expressed in reduction in viral titres or log10 of viral titres. Standard deviations must be provided and also the number of repetition (in figures: error bars are not explained: standard deviation, standard error of mean, or… ?
  • Line57: in the mat &met, polymer 265L should be explained (I confess I do not know what is 265L and I cannot find it by an internet search); the NH4 quaternary compounds labelled 3310 and 1297 have to be more detailed;
  • Result section: the limit of detection of your method must be given;
  • Line65: provide the full name of DDMAB;
  • Line66: “additional combinations”: it does not mean anything in this results section: to be deleted or the name of these combinations should be given;
  • Line82: what does precisely mean “complete inactivation”? It should be inactivation under the limit of detection and reported as a reduction in viral titres. Ususally a 3 to 4 log10 reduction in viral titres is considered as efficacious;
  • Figure 2: citric acid as positive control is not explained in the mat&meth;
  • Figures 1 and 2: the scale of the y-axis should be harmonised: it ends at 0, 1 or 10 PFU; the unit should be explained: PFU/which volume?
  • Discussion: two points deserve discussion: the persistence of the virucidal coating on the mask and the possible reuse of such treated masks;
  • Line136: it should be appropriate to give a short list of these pathogens that could be inactivated by NH4 quaternary compounds, especially non enveloped viruses;
  • 3 ref seem a bit short, even for a short communication.

Author Response

We thank the reviewers for their helpful comments and suggested revisions.  We have replied to these concerns in italics below.

Reviewer #1:

The testing of virucidal activity of decontamination methods against SARS-CoV-2 is timely and, here, even more with a virucidal pretreatment of masks. This short paper addresses these needs. It investigates the inactivation capacity of the coating of surgical masks with quaternary ammonium compounds. A virucidal effect is evidenced against SARS-CoV-2 and apparently there is no doubt about this result. However, acceptance of this article should be subject to a more precise description of the methods and a correct presentation of the results, as explained in the following comments. Indeed an elementary rule of a primary scientific publication is that results can be reproduced. This paper does not give enough information pertaining the reproduction of the work by another laboratory:

  • Abstract: SARS-CoV-2 instead of “SARs-coV-2”

This has been changed, line 11

  • Abstract: “inactivates a high dose of …”: please be more precise in this statement even if it is in the summary

We have added the upper limit of virus tested in this paper on line 14.

  • Line24: “for days”: be more precise

We included the range of days listed in the studies on lines 24-25.

  • Line33: “high titers of live SARS-CoV-2”: be more precise about SARS-CoV-2 titres

We have included the titer of the viral inoculum (1 x 105 pfu) in this sentence, now line 34.

  • Line37: surgical masks: which brand? Is it a regular officially approved surgical mask?

We have indicated the type of surgical mask and that it is an approved Class I medical device on lines 38-39

  • Line 39: what is the composition of the disinfectant chemical embedded on the outside mask layer? (There is some information in the result section but it should appear in this section)

We have included this information on lines 41-47.

  • Line46: how many mask pieces have been used? How many replicates in this trial? (Some information can be found in the figures but it needs to be stated in this section)

This has been updated in lines 55-57.

  • Line46: How masks have been inoculated? Deposit of droplets? Which volume, etc. In which medium was the virus resuspended?

This information was added in lines 60-63.

  • Line46: provide a reference regarding the pfu method and the cell line used (Vero E6 cells)

We have added this information in lines 68-70, including references for the method as previously published by our group.

  • Line 48: give the indicated times (even if they are found somewhere in the results)

We have added “for the indicated times” on line 60 since the incubation times differed between the different experiments.

  • Line49: extraction simply by incubation, without any agitation?

This is correct.

  • Viral reduction? Comparison is not a result: it should be expressed in reduction in viral titres or log10 of viral titres. Standard deviations must be provided and also the number of repetition (in figures: error bars are not explained: standard deviation, standard error of mean, or… ?

We have added the reduction in viral titers to the text in line 88.  We prefer showing raw data in the figures, as this allows the reader to note that complete inactivation was achieved (rather than only list fold reduction of viral titers, which can only be listed as >105 reduction and therefore isn’t quantitative).  Our figure presentation is consistent with data presentation in other SARS-CoV-2 inactivation manuscripts (such as Riddell et al., The effect of temperature on persistence of SARS-CoV-2 on common surfaces. Virol J. 2020 Oct 7;17(1):145; and van Doremalen et al, Aerosol and Surface Stability of SARS-CoV-2 as Compared with SARS-CoV-1. N Engl J Med. 2020 Apr 16;382(16):1564-7.)

The figure legends already list the N and repetitions, and we have added the information for error bars in the figure legends.

  • Line57: in the mat &met, polymer 265L should be explained (I confess I do not know what is 265L and I cannot find it by an internet search); the NH4 quaternary compounds labelled 3310 and 1297 have to be more detailed

We have modified the methods in lines 41-44 to explain that 265L is a free-radical, polymer film commercially produced by Green Theme Technologies.  It is an acrylate-type polymer that is thermally stable and fluorocarbon-free. “3310 and 1297” are short terms for two different quats that can be found in the patent application by Green Theme Technologies once the patent is published.

  • Result section: the limit of detection of your method must be given.

This has been added in lines 69-70.  .

  • Line65: provide the full name of DDMAB;

DDMAB is an accepted chemical abbreviation for a water-soluble quat.  Its full chemical name can be found in the patent application by Green Theme Technologies when the patent application is published.

  • Line66: “additional combinations”: it does not mean anything in this results section: to be deleted or the name of these combinations should be given;

This has been modified in lines 85-86 to include the description of the combination.

  • Line82: what does precisely mean “complete inactivation”? It should be inactivation under the limit of detection and reported as a reduction in viral titres. Ususally a 3 to 4 log10 reduction in viral titres is considered as efficacious;

Complete inactivation means no live virus was detected.  As mentioned above, we have added some additional information for partial inactivation based on log10 reduction.

  • Figure 2: citric acid as positive control is not explained in the mat&meth;

We have modified the methods to include “natural products” on lines 46-47.

  • Figures 1 and 2: the scale of the y-axis should be harmonised: it ends at 0, 1 or 10 PFU; the unit should be explained: PFU/which volume?

We thank the reviewer for bringing this oversight to our attention.  We have modified the graphs consistently show the data as PFU on a log scale and have expanded the results text to more clearly explain the data.  We have also modified the figure legends to explain that “PFU” on the Y axis is defined by calculated total PFU for the entire sample.

  • Discussion: two points deserve discussion: the persistence of the virucidal coating on the mask and the possible reuse of such treated masks;

We have included information on this topic in the discussion on lines 156-159. 

  • Line136: it should be appropriate to give a short list of these pathogens that could be inactivated by NH4 quaternary compounds, especially non enveloped viruses;

We have added text and references to the discussion in lines 161-163 to address this concern. Quaternary amines are widely used and approved by the EPA for disinfectant sprays against bacterial, viruses and fungi.  A quite common disinfectant spray that contains a water-soluble quat is Lysol disinfectant spray. According to the manufacturer information, this product is approved by the EPA against the following viruses: Viruses: Avian Influenza A Virus (H1N1), Influenza A Virus (New Caledonia/20/99), Influenza B Virus (Strain B/Hong Kong/5/72), Rhinovirus Type 39, Feline calicivirus (Norovirus), Rotavirus WA, Herpes Simplex Virus Type 1, Herpes Simplex Virus Type 2, Respiratory Syncytial Virus (RSV).

  • 3 ref seem a bit short, even for a short communication.

We have added additional references.

Reviewer 2 Report

While this paper is very interesting and a potentially good paper, I don't think that this paper is appropriate as a short communication. A short communication is only appropriate for papers that attempt to address a highly specific question , not one that attempt to address broad questions. This paper falls into the second category,  Therefore, when the authors attempt to write this as a short communication, it raises more questions than provides answer. A real problem is that there are more complex issues involved  and they are being swept under the carpet. I will leave the decision regarding whether this paper should continue as a Communication but I do believe more details should be added particularly to the issues put forth below:

1)The mechanisms of antimicrobial actions  have not been mentioned. I suggest a table that list the various antiseptics with a column that list out the possible  mechanism of action. If there is enough room, the chemical structures of the compounds would be good:

https://www.ncbi.nlm.nih.gov/pmc/articles/PMC6100501/?fbclid=IwAR1lkh6_ZbuVlfBK-suOtjDQwdvvju3--OSvo3z7BFHbNnxEsMNIy0orp5s

2) The topic of using antimicrobial compounds raises a much more complex issues especially pertaining to the nature of SARS-CoV-2. Goh et al reported that SARS-CoV-2 has one of the hardest ourter shell (M) in all of CoV because of its evolutionary association with a burrowing animal, pangolin, that comes in contact with buried feces. Because of the virus's hard outer shell it is more capable of resisting antimicrobial enzymes found in the mouth and mucus. As a result, the virus shed tremendous amount of infectious particles in patients. This particular issue is tricky as it implies that many antiseptics may not work against SARS-CoV-2  and indeed the results presented  do show that some of the compounds are useless against  the virus.

https://pubmed.ncbi.nlm.nih.gov/32244041/

https://pubmed.ncbi.nlm.nih.gov/32790362/

https://pubmed.ncbi.nlm.nih.gov/21441478/

https://pubmed.ncbi.nlm.nih.gov/10377100/

3) A discussion to address (1) and (2) would be appropriste. Attempts to interpolate or , if not, extrapolate the mechanisms of action of the antiseptics against SARS-CoV-2 and the reasons for failure/success in deactivation of the virus would be good

Author Response

We thank the reviewers for their helpful comments and suggested revisions.  We have replied to these concerns in italics below.

Reviewer #2

While this paper is very interesting and a potentially good paper, I don't think that this paper is appropriate as a short communication. A short communication is only appropriate for papers that attempt to address a highly specific question , not one that attempt to address broad questions. This paper falls into the second category,  Therefore, when the authors attempt to write this as a short communication, it raises more questions than provides answer. A real problem is that there are more complex issues involved  and they are being swept under the carpet. I will leave the decision regarding whether this paper should continue as a Communication but I do believe more details should be added particularly to the issues put forth below:

1)The mechanisms of antimicrobial actions  have not been mentioned. I suggest a table that list the various antiseptics with a column that list out the possible  mechanism of action. If there is enough room, the chemical structures of the compounds would be good:

https://www.ncbi.nlm.nih.gov/pmc/articles/PMC6100501/?fbclid=IwAR1lkh6_ZbuVlfBK-suOtjDQwdvvju3--OSvo3z7BFHbNnxEsMNIy0orp5s

It is thought that quaternary ammonium salts inactivate enveloped viruses by disrupting the viral lipid envelope (PMID 25362069).  We have added this information to the discussion on lines 128-130.  We are unable to provide the chemical structures of the compounds at this point, given that the patent is pending on these compounds.

2) The topic of using antimicrobial compounds raises a much more complex issues especially pertaining to the nature of SARS-CoV-2. Goh et al reported that SARS-CoV-2 has one of the hardest ourter shell (M) in all of CoV because of its evolutionary association with a burrowing animal, pangolin, that comes in contact with buried feces. Because of the virus's hard outer shell it is more capable of resisting antimicrobial enzymes found in the mouth and mucus. As a result, the virus shed tremendous amount of infectious particles in patients. This particular issue is tricky as it implies that many antiseptics may not work against SARS-CoV-2 and indeed the results presented  do show that some of the compounds are useless against  the virus.

https://pubmed.ncbi.nlm.nih.gov/32244041/

https://pubmed.ncbi.nlm.nih.gov/32790362/

https://pubmed.ncbi.nlm.nih.gov/21441478/

https://pubmed.ncbi.nlm.nih.gov/10377100/

We thank the reviewer for this interesting perspective.  Fortunately, standard laboratory disinfectants work well against SARS-CoV-2 (quaternary ammonium, peroxide-based products, bleach, and alcohol-based products).  It is possible that less potent disinfectants may not be as effective against SARS-CoV-2.  We have added to the discussion to address this point in lines 128-139.

3) A discussion to address (1) and (2) would be appropriate. Attempts to interpolate or, if not, extrapolate the mechanisms of action of the antiseptics against SARS-CoV-2 and the reasons for failure/success in deactivation of the virus would be good

We have added information to the discussion as described above.

Round 2

Reviewer 1 Report

Accepted in its present form: the manuscript was correctly modified.

Reviewer 2 Report

Improvements made.